# Comparative Analysis of Spectral Broadening Techniques for Optical Temperature Sensing in Yttrium Fluoride (YF_3_) Doped with Neodymium

**DOI:** 10.3390/s25072324

**Published:** 2025-04-06

**Authors:** Ruan P. R. Moura, Bárbara M. Cruz, Tatiane S. Lilge, Adriano B. Andrade, Mario E. G. Valerio, Zélia S. Macedo, José J. Rodrigues, Márcio A. R. C. Alencar

**Affiliations:** 1Physics Department, Federal University of Sergipe, São Cristovão 49107-230, Brazil; ruan07@academico.ufs.br (R.P.R.M.); mcruzbarbara@gmail.com (B.M.C.); tatianelilge@academico.ufs.br (T.S.L.); abandrade@academico.ufs.br (A.B.A.); megvalerio@academico.ufs.br (M.E.G.V.); zelia.macedo@academico.ufs.br (Z.S.M.);; 2Laboratory of Corrosion and Nanotechnology (LCNT), Federal University of Sergipe, São Cristovão 49107-230, Brazil; 3Graduate Program in Materials Science and Engineering, Federal University of Sergipe, São Cristovão 49107-230, Brazil

**Keywords:** optical temperature sensor, valley-to-peak intensity ratio, bandwidth, biological window

## Abstract

In this work, YF_3_:Nd^3+^ powder was synthesized using the microwave-assisted hydrothermal method at a low temperature (140 °C) and short synthesis time (1 h). The photoluminescence and optical temperature sensing properties of YF_3_:Nd^3+^ were examined using 800 nm laser excitation, focusing on the emission corresponding to the ^4^F_3/2_ → ^4^I_9/2_ transition of Nd^3+^. The performance of YF_3_:Nd^3+^ as an optical temperature sensor was evaluated using the full width at half maximum (FWHM), band broadening at 30% of maximum intensity (Δλ_30%_), and valley-to-peak intensity ratio (VPR) techniques. All techniques demonstrated good repeatability and reproducibility. The best results were obtained using the VPR (V1/P1) method, which exhibited the highest relative sensitivity and the lowest temperature uncertainty, with values of 0.69 ± 0.02% K^−1^ and 0.46 ± 0.09 K at 303 K, respectively. YF_3_:Nd^3+^ shows promise as an optical temperature sensor operating entirely within the first biological window.

## 1. Introduction

Temperature is a fundamental thermodynamic quantity, playing a crucial role in various technological and scientific fields [1]. With the constant advances in these areas, the demand for temperature sensors with high precision, spatial resolution, and fast response time is increasingly necessary, especially in challenging environments where direct contact is not viable [2,3].

In the biological field, for instance, the application of optical temperature sensors is particularly challenging for in vivo measurements due to complex structures in human and animal bodies [4]. These optical thermometers must possess high thermal, chemical, and mechanical stability, low cytotoxicity, and operate within the so-called biological windows (BWs) for deep tissue measurements [5,6]. The biological windows are wavelength regions of the electromagnetic spectrum in which light absorption and scattering by biological tissue are minimized [7,8,9]. Currently, four BWs were identified and exploited for optical thermometry: the spectral region from 650 to 950 nm, defined as the first BW (I-BW); the second BW (II-BW) from 1000 to 1350 nm; the third BW (III-BW) from 1400 to 2000 nm; and the fourth BW (IV-BW) centered at 2200 nm [7,8,9]. Advancements in micro- and nanosized optical temperature sensors based on inorganic hosts doped with lanthanides have shown promising potential in meeting these requirements. Particularly, sensors based on solid hosts doped with neodymium (Nd^3+^) have the capability to be excited and emit light within the biological windows, which allows for more accurate temperature measurements in deeper parts of the biological tissue [10,11,12].

These optical temperature sensors can be developed using a wide variety of optical parameters that depend on temperature, such as the luminescence intensity ratio (LIR), lifetime, line shift, bandwidth, etc. [13,14]. The LIR is the most widely used method in optical thermometry due to its high precision and immunity to external interference, excitation source fluctuations, and energy loss [15,16,17,18,19].

On the other hand, bandwidth methods, although less utilized, offer precision and stability, and are also insensitive to laser power fluctuations [15,20]. In those, the change on the spectral width of the luminescence band with temperature is exploited. Usually, these methodologies are based on the variation in the bandwidth measured at half maximum of the luminescence intensity, or the full width at half maximum (FWHM). Although generally associated with lower relative sensitivity [21], the research published by Rodríguez et al. in 2018 [22] using the full width at half maximum (FWHM) as the thermometric parameter, and Y_(1−x)_Nd_x_AlO_3_ (YAP):Nd^3+^ as the luminescent probe, achieved a relative sensitivity of 3.3% K^−1^ and a low temperature uncertainty (δT) of 0.37 K. This demonstrated that good performance of the optical temperature sensor can be obtained with bandwidth-based methodologies.

An alternative bandwidth strategy was proposed by Laia et al. [23]. In this work, they characterized the temperature-dependent luminescence in I-BW of LiBaPO_4_:Nd^3+^ microcrystals, excited in I-BW. They observed significant bandwidth broadening, but the overlap amongst the observed emission lines prevented the evaluation of the FWHM. To overcome this limitation, the authors defined the quantity width measured at 90% of the maximum intensity, named Δλ_90%_, and analyzed its behavior with temperature. They showed that, for this material, the temperature-sensing scheme based on Δλ_90%_ was more sensitive than the commonly used FWHM technique and allowed for a more comprehensive analysis of the emission spectrum.

Another promising method is the valley-to-peak intensity ratio (VPR), developed by Yuan Zhou et al. in 2015 [24]. In this method, the ratio of the valley intensity, formed by the partial overlap of two adjacent peaks, to one of the intensities of the peaks forming the valley is calculated. Zhou et al. [25] used the VPR method to develop an optical temperature sensor employing ZrO_2_:Er^3+^/Yb^3+^ as the sensing probe. They performed ex vivo temperature monitoring across the three biological windows. This research achieved high relative sensitivity in all biological windows, with the highest being 1.3% K^−1^ found in III-BW.

All these bandwidth methods are strongly dependent on the dopant and host chemical and physical properties. Because of this, it is difficult to perform a comprehensive comparison amongst their performances, besides the evaluation of the relative sensitivity of the sensor scheme. Here, a temperature-sensing methodology based on the bandwidth broadening measured at 30% of the maximum intensity (Δλ_30%_) is proposed, operating in the first biological window, using neodymium-doped yttrium fluoride (YF_3_:1% Nd^3+^) nanocrystals as sensing probes. A comparative analysis was performed between the VPR method and the FWHM and Δλ_30%_ methods. This comparison aims to evaluate the efficiency of single-band broadening techniques (FWHM and Δλ_30%_) versus two-band broadening (VPR), considering the need for accurate, non-invasive thermal readings within the first biological window.

## 2. Materials and Methods

### 2.1. Synthesis and Structural Characterization of YF_3_:1%Nd

Yttrium fluoride (YF_3_) polycrystalline samples were synthesized using a microwave-assisted hydrothermal technique (MAH), as described in detail in ref. [26]. Briefly, the initial reactants were Y(NO_3_)_3_·6H_2_O (Alfa Aesar, Ward Hill, MA, USA, with 99.9% purity) and NH_4_F (Neon, São Paulo, Brazil, with 99.9% purity) in their stoichiometric ratios. To prepare Nd-doped YF_3_, 1 mol% of Nd(NO_3_)_3_·6H_2_O (Sigma-Aldrich, Steinhein, Germany, with 99.9% purity) was used to replace an equivalent amount of yttrium nitrate. Each precursor was separately dissolved in deionized water and stirred magnetically for 60 min. Afterward, the NH_4_F solution was slowly added to the Y(NO_3_)_3_ solution while maintaining continuous stirring. The mixture was stirred for another 30 min and then transferred to a Teflon vessel, which was subsequently filled with deionized water up to a total volume of 100 mL. The sealed Teflon vessel was placed in a microwave apparatus (Brazilian patent 2008-PI0801233-4) operating at 2.45 GHz with a maximum power output of 800 W. The temperature inside the vessel gradually increased to 140 °C at a rate of 10 °C per minute and held steady for 60 min. Following the reaction, the resulting YF_3_ powders were thoroughly washed and centrifuged with deionized water until the pH was neutral. The powders were then dried on a hotplate at 90 °C.

The characterization of the structure was made by X-ray diffraction (XRD) analyses, conducted using a Panalytical EMPYREAN (Malvern Panalytical, Almelo, The Netherlands) diffractometer configured with Bragg–Brentano geometry. The X-ray source utilized Cu Kα radiation, with an operating voltage of 40 kV and a current of 40 mA. Data acquisition was performed in steps of 0.026° with a time of 200 s per step.

Microstructural characterization of the samples was carried out using Scanning Electron Microscopy (SEM) on a JEOL JSM-6510LV (JEOL, Tokio, Japan), operating at 15 kV. Particle size distribution was determined from SEM images utilizing the ImageJ software (version 1.52a) for analysis.

### 2.2. Luminescence Experimental Setup

The luminescence experimental setup is presented schematically in Figure 1. An 800 nm CW laser, with 6 mW power, was used to excite the sample. The yttrium fluoride powder was placed on an aluminum sample holder, occupying a circular area with a diameter of 4 mm. Subsequently, the sample was placed on a heating plate and subjected to heating, with temperature control carried out by a thermocouple that measured the temperature near the sample. A polarizer and an FD 800-10 filter (Thorlabs, Newton, NJ, USA) were placed before the sample to control the incident light power and eliminate spectral noise from the laser, respectively. A set of mirrors directed the laser light, and a converging lens with a focal length of 10 cm focused the laser light on the sample. The emitted light passed through two converging lenses before being collected by an optical fiber connected to the Maya 2000 spectrometer (Ocean Optics, Orlando, FL, USA) which allowed for the analysis of the sample’s luminescent signal. A set of FGL-850 and FEL-850 filters (Thorlabs) were employed to remove any signal contamination from the excitation laser to the luminescence spectrum.

## 3. Results

### 3.1. Structural and Morphological Properties

The diffraction patterns of the produced undoped YF3 and the YF3:1%Nd^3+^ samples are essentially the same as reported in our previous work [26]. They indicate that the samples are crystalline with a single phase corresponding to the orthorhombic crystal structure with space group Pnma symmetry [$D_{2h}^{16}$], matching the ICSD pattern N° 26,595.

The SEM images are presented in Figure 2a,b. In Figure 2a, the images show that the particles have a rod-like shape and are well dispersed. In Figure 2b, the magnified image provides a clearer view of how the particles are organized, as well as their shape. The particles have an average size of 644 ± 127 nm in length and 344 ± 47 nm in width.

### 3.2. Optical Characterization and Sensing Analyses

Figure 3a shows the emission spectrum of YF_3_:1% Nd^3+^, excited with an 800 nm continuous wave (CW) laser, measured at room temperature. Two emission bands in the infrared region can be observed. One centered at 884 nm, which is associated with the radiative decay of electrons from the ^4^F_3/2_ level of neodymium to the ground state ^4^I_9/2_. The second one is centered around 1056 nm, resulting from the decay from the ^4^F_3/2_ level to the excited state ^4^I_11/2_. A simplified energy level diagram is presented in Figure 3b, illustrating the radiative and non-radiative processes of neodymium. In this figure, it can be observed that neodymium electrons are excited by the 800 nm CW laser, transitioning from the ground state ^4^I_9/2_ to the higher energy state ^4^F_5/2_. From this level, the ions decay non-radiatively to the ^4^F_3/2_ level. Subsequently, these ions can decay radiatively either to the ground state, resulting in an emission band located in the first biological window, or to the first excited state ^4^I_11/2_, generating an emission band in the second biological window. Moreover, due to the crystalline field of YF_3_, the energy levels of Nd^3+^ split into Stark’s sublevels and the emission band is composed by set of narrows emission peaks associated with the transitions between the sublevels of the ^4^F_3/2_ excited state and the sublevels of ^4^I_9/2_ and ^4^I_11/2_.

To evaluate the potential use of YF_3_:1% Nd^3+^ as a temperature sensor, temperature variations were performed to observe the behavior of the material’s spectrum. The results are shown in Figure 4a. In this figure, the emission band at I-BW, measured at different temperatures, varying from 303 K to 373 K, is presented. The overall behavior observed is a typical temperature quenching of the luminescence. Nevertheless, the intensities associated with the individual transitions between the Stark’s sublevels present a richer behavior. In our previous work [26], the behavior of the ^4^F_3/2_ → ^4^I_9/2_ emission band of YF_3_:1% Nd^3+^ with temperature variations and the transitions of the Stark’s sublevels is detailed. Indeed, by properly exploiting the temperature behavior of these emission lines, a LIR-based temperature sensor was proposed and characterized.

To further highlight the luminescent variations in the emission band corresponding to the ^4^F_3/2_ → ^4^I_9/2_ transition, the data were normalized with respect to peak with the highest intensity, labeled as P_1_, as shown in Figure 4b. It is noticeable that all the emission peaks broaden as the temperature increases. To analyze the effect of peak broadening for use in optical temperature sensing, the peak broadening technique was assessed on all peaks. To quantify its width using FWHM, we determine directly from the spectrum the wavelengths at which the corresponding intensities are equal to half of the peak intensity. Hence, the FWHM is obtained simply by the difference between these two wavelengths, as shown in Figure 4c. Nevertheless, only peak P_1_ showed sufficient broadening, using FWHM to quantify the spectral width, to allow the use of the band broadening technique. This is observed in detail in Figure 4c, where the P_1_ spectra measured at 303 and 373 K are shown.

We also defined the width of the band measured at 30% of the peak intensity, labeled as ${\Delta \lambda }_{30\%}$. This value is also obtained directly from the spectrum. First, we identify the wavelengths at which the corresponding intensities are equal to 30% of the peak intensity. Then, ${\Delta \lambda }_{30\%}$ is the diference between these two wavelength values. In Figure 4c, the width measured at half of the maximum intensity (FWHM) and the width measured at 30% of the peak intensity are also highlighted. It is possible to observe that the broadening at 30% of the maximum intensity of peak P_1_ is greater than the FWHM broadening. This suggests that performing temperature sensing by monitoring the width at 30% could provide an improvement in the sensitivity obtained in comparison with the FWHM method.

Hence, we characterized both width definitions as the thermometric parameter of a temperature sensor based on bandwidth. The results are shown in Figure 5a. As can be seen, both methods exhibit linear behavior and were fitted according to the equation below.(1)$${\Delta \lambda }_{i}=c_{i}+d_{i}T,$$
where ${\Delta \lambda }_{i}$ is the bandwidth measured at 50% of the peak maximum (i=FWHM) or at 30% of the maximum (${i=\Delta \lambda }_{30\%})$, T is the temperature, and $c_{i}$ and $d_{i}$ are empirical constants.

Thermal sensitivity is one of the most important performance parameters in optical thermometry. It is defined as the rate of change in the analyzed thermometric parameter (Q) in relation to the temperature (Sa = dQ/dT). For direct comparison between various types of optical temperature sensors, relative sensitivity (Sr = Sa/Q) is used, since it is independent of the experimental setup and the nature of the material [27,28]. The relative sensitivities of the methods presented above were calculated using the following equation:(2)$$S_{r}^{{\Delta \lambda }_{i}}=\frac{1}{{\Delta \lambda }_{i}}\frac{d({\Delta \lambda }_{i})}{dT}=\frac{d_{i}}{{\Delta \lambda }_{i}},$$

The results of the relative sensitivities for the two bandwidth broadening techniques are shown in Figure 5b. It can be observed that the ${\Delta \lambda }_{30\%}$ technique presented a higher relative sensitivity than the traditional FWHM broadening technique. This was expected since the observed bandwidth broadening at 30% of P_1_ intensity was greater than at 50% intensity. In comparison with the relative sensitivity obtained using the LIR method [26], the ${\Delta \lambda }_{30\%}$ technique also exhibited superior performance, while the FWHM is the least sensitive method.

It is worth noting that the emission spectrum of YF_3_:Nd^3+^, illustrated in Figure 4b, also exhibits all the necessary characteristics for implementing the VPR technique. These characteristics are as follows: (i) the presence of two peaks partially overlapped; (ii) each spectral line conforms to a Lorentzian profile; and (iii) the peaks broaden homogeneously with increasing temperature. Figure 4b highlights six points with potential use for VPR in optical temperature sensing. Peaks P_1_, P_2_, and P_3_ are located at 866.8, 894.2, and 901.3 nm, respectively, while valleys V_1_, V_2_, and V_3_ are at 861.1, 886.7, and 896.9 nm. It was observed that as the temperature increases, the valley intensities also increase. This process occurs due to the broadening of the peaks as the temperature increases, resulting in greater overlap between them and consequently, an increase in the intensity of the valley [24]. All possible valley-to-peak ratios (V_1_/P_1_, V_2_/P_2_, V_3_/P_3_) were evaluated. As shown in Figure 6a, the VPR technique exhibits a linear dependence on temperature variation and was thus fitted using the equation below.(3)VPR=a+bT,
where a and b are material-specific constants. The relative sensitivity using this technique was evaluated using Equation (4).(4)$$S_{r}^{VPR}=\frac{1}{VPR}\frac{d(VPR)}{dT}=\frac{b}{VPR},$$

In Figure 6b, the results of relative sensitivity for all the tested VPR are shown. The best performance was obtained for the V_1_/P_1_ ratio, with a sensitivity of 0.69% K^−1^. This was the highest relative sensitivity value measured amongst all the analyzed techniques in this work and in ref. [26].

In addition to relative sensitivity, other characteristics of an optical temperature sensor were analyzed: material repeatability, reproducibility, and temperature uncertainty [29,30]. These properties are essential for evaluating the sensor’s accuracy and reliability. Temperature uncertainty corresponds to the smallest temperature change that can be experimentally detected and was calculated using Equation (5).(5)$$\delta T=\frac{1}{S_{r}}\frac{\delta Q}{Q},$$
where δQ/Q is the relative uncertainty in determining the thermometric parameter. It is important to note that the temperature uncertainty strongly depends on the experimental setup used, the measured conditions, and the signal-to-noise ratio. The temperature uncertainty can be improved, for instance, by using a system with high relative sensitivity and/or by improving the signal-to-noise ratio of the material’s emission spectrum. While the former is achieved with the proper choice of the probe material and thermometric parameter; the latter can be carried out by increasing the integration time or the number of measurements in the acquisition of the emission spectrum [2,10,28]. Regardless, it is the temperature uncertainty parameter that provides a more reliable measure of the material’s thermal resolution.

The temperature uncertainties obtained using the band broadening and VPR methods are presented in Figure 7. Except for the temperature uncertainty obtained using the V_2_/P_2_ ratio, the uncertainties determined by the other methods yielded excellent results, with values below 1 K within the physiological temperature range (303–333 K). This indicates high reliability in temperature measurement, making it highly promising for applications in biological environments. The methodologies based on the bandwidth measured at 30% of the peak (Δλ_30_%) and the valley–peak intensity ratio V_1_/P_1_ presented the best results for temperature uncertainty among all tested techniques, varying from 0.50 to 0.68 and from 0.46 to 0.69 K, respectively, within the investigated temperature range. These results are even better than what was obtained using Stark’s transition-based LIR methodology with the same material and experimental setup [26], which varied from 0.70 to 1.10 K.

Reproducibility is defined as the precision of temperature measurements under varying experimental parameters. In Figure 8, the results of the thermometric parameters obtained in four different temperature scans are presented. These scans were performed on different days and with distinct samples of the produced YF_3_:1% Nd^3+^ powder. Using this approach, we verified that all techniques demonstrate good reproducibility, as can be observed in the figures. The fluctuation of the measured values amongst the distinct scans can be quantified, for instance, by calculating the sample standard deviation of each measured point. We observed that the sample standard deviations were smaller than 6.13%.

The repeatability (R) of a thermometer can be defined as the device’s ability to consistently provide the same result in repeated measurements under identical conditions. To assess this property, ten consecutive heating and cooling cycles of the sample were conducted. The results are shown in Figure 9. Repeatability can be quantified using Equation (6).(6)$$R_{\%}=\left[1-\frac{max\left(\left|Q_{m}-Q_{c}\right|\right)}{Q_{m}}\right]\times 100\%,$$
where $Q_{m}$ is the average value of the thermometric parameter and $Q_{c}$ is the value of the thermometric parameter from the measurements taken during the heating and cooling cycles. As can be seen in Figure 9, all the tested techniques exhibit good repeatability, varying from 97% to 99.16%. The results show that the values measured by the VPR, FWHM, and Δλ_30%_ techniques lie within the confidence interval of 10 times the sample standard deviation of the mean, which is marked by the blue background. This indicates that all methodologies present good reliability and consistency in temperature measurements.

In Table 1, a comparison of the thermometric performance of different rare-Earth-doped optical temperature sensors is presented, utilizing the FWHM, VPR, and LIR techniques. Amongst the systems exploiting FWHM, YF_4_:Nd investigated in this work presents a better performance than YVO_4_:Nd^3+^ [31], YVO_4_:Eu^3+^ [32], and CaGdAlO_4_:Tm^3+^/Yb^3+^ [33], for instance, with slightly higher relative sensitivity and lower temperature uncertainty. The Δλ_30%_ technique shows relative sensitivity about twice as high as traditional band broadening techniques with a substantial improvement in temperature uncertainty. Among all the methods investigated, the V1/P1 technique stands out. This methodology exhibited the best relative sensitivity, and the lowest temperature uncertainty of the sensing strategies investigated in this work, even better than using the LIR method [26], indicating high precision and reliability in temperature detection. Finally, compared with other sensing schemes and materials [31,32,33,34,35,36,37,38,39,40,41,42], this system exhibited competitive values of relative sensitivity (>0.6% K^−1^) and temperature uncertainty (<0.5 K). In comparison with Nd-doped YAP [22], although YF_3_ exhibited a lower sensitivity value, the temperature uncertainties of both systems were of the same order of magnitude. Moreover, the Nd-doped YF_3_ sensing schemes operate entirely in the infrared region (the first biological window), which can be an advantage for several applications, such as deep tissue measurements.

## 4. Conclusions

In summary, rod-like submicrometric YF_3_:Nd^3+^ nanocrystalline particles were synthesized and their performance as optical temperature probes operating within the first biological window was evaluated. Three different thermometric methodologies based on temperature-dependent spectral broadening, namely VPR, FWHM, and Δλ_30%_, were compared in the temperature range of 303–373 K. Reproducibility, repeatability, relative sensitivity, and temperature uncertainty were analyzed to ensure the accuracy and reliability of the results. All techniques demonstrated good reproducibility and repeatability. The Δλ_30%_ technique showed improved relative sensitivity and temperature uncertainty in comparison to the FWHM technique. Additionally, when compared to other studies that utilized traditional band broadening techniques, Δλ_30%_ exhibited about twice the relative sensitivity and provided a significant improvement in temperature uncertainty. The V_1_/P_1_ method presented the highest relative sensitivity and the lowest temperature uncertainty, with values of 0.69 ± 0.02% K^−1^ and 0.46 ± 0.09 K at 303 K, respectively. Overall, the thermometric performance of YF_3_:Nd^3+^ powder offers a good option for optical temperature sensing operating entirely within the first biological window.

## Figures and Tables

**Figure 1 sensors-25-02324-f001:**
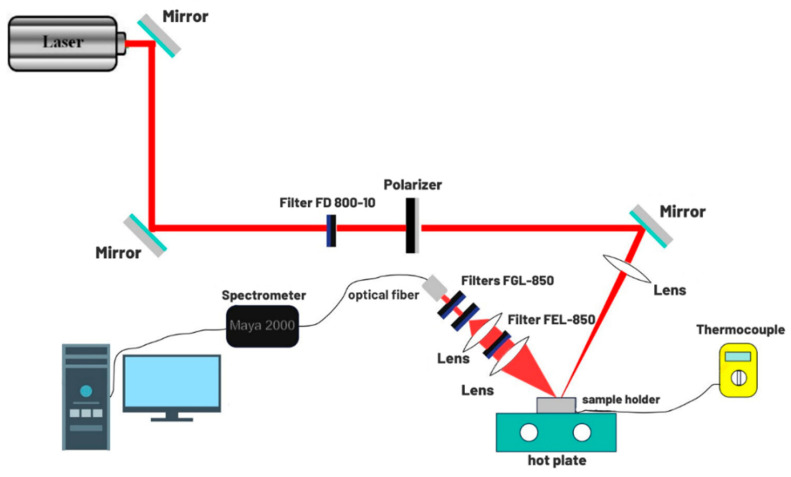
The experimental setup of the thermometric calibration of the YF_3_:1% Nd^3+^.

**Figure 2 sensors-25-02324-f002:**
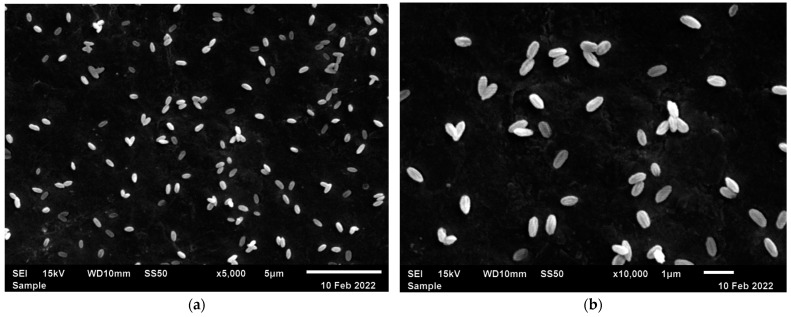
(**a**) SEM images of YF_3_, with magnification of ×5000; (**b**) SEM images of YF_3_, with magnification of ×10,000.

**Figure 3 sensors-25-02324-f003:**
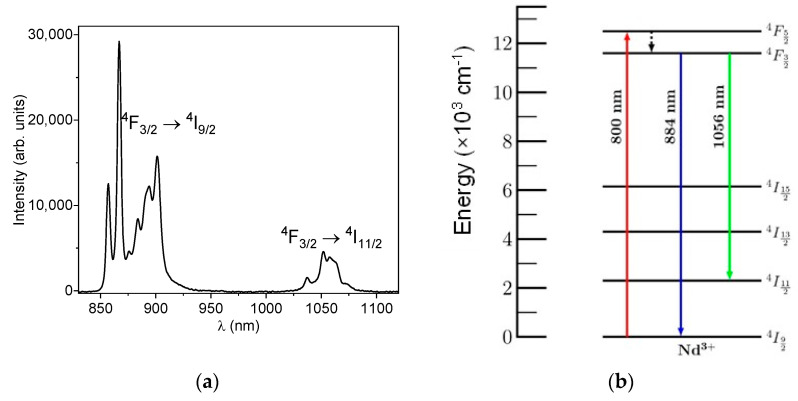
(**a**) Emission spectrum of YF_3_:1% Nd^3+^; (**b**) energy level diagram of Nd^3+^ in YF_3_. The solid arrows correspond to the radiative transitions and the dashed arrow represents the non-radiative transition.

**Figure 4 sensors-25-02324-f004:**
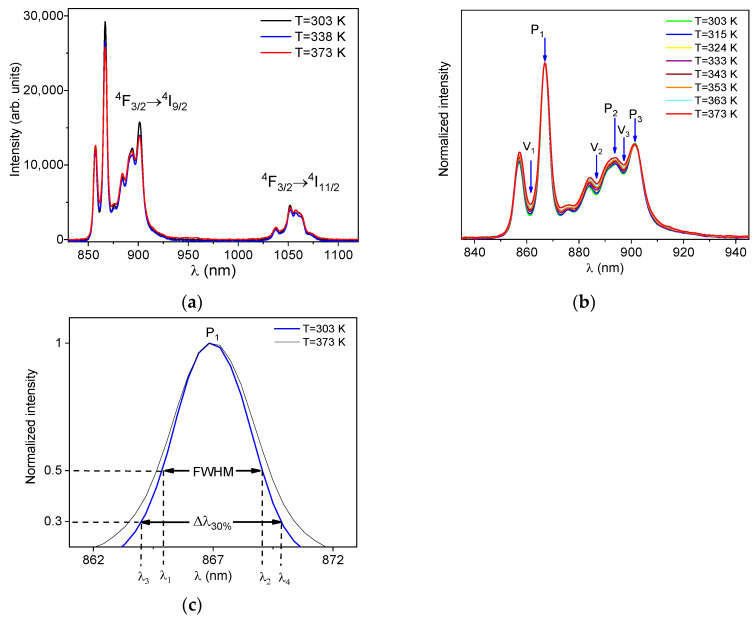
(**a**) The emission spectra of the ^4^F_3/2_ → ^4^I_9/2_ transition, measured at three different temperatures. (**b**) The normalized emission spectra of band ^4^F_3/2_ → ^4^I_9/2_ at different temperatures. (**c**) Peak P_1_ measured at 303 and 373 K, and the definition of the width measured at half of the maximum intensity ($FWHM=\lambda_{2}-\lambda_{1}$) and measured at 30% of the maximum intensity (${\Delta \lambda }_{30\%}=\lambda_{4}-\lambda_{3}$).

**Figure 5 sensors-25-02324-f005:**
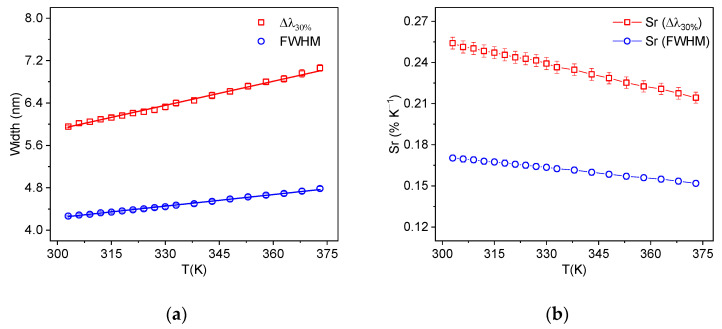
(**a**) Change in the bandwidth; (**b**) relative sensitivity with temperature using bandwidth techniques. Solid lines in (**a**) correspond to fit using Equation (1).

**Figure 6 sensors-25-02324-f006:**
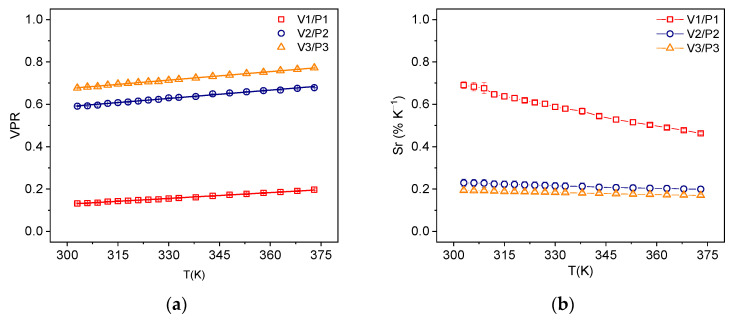
(**a**) VPR; (**b**) $S_{r}^{\mathrm{VPR}}$ as function of temperature for three valley–peak ratios YF_3_:Nd^3+^.

**Figure 7 sensors-25-02324-f007:**
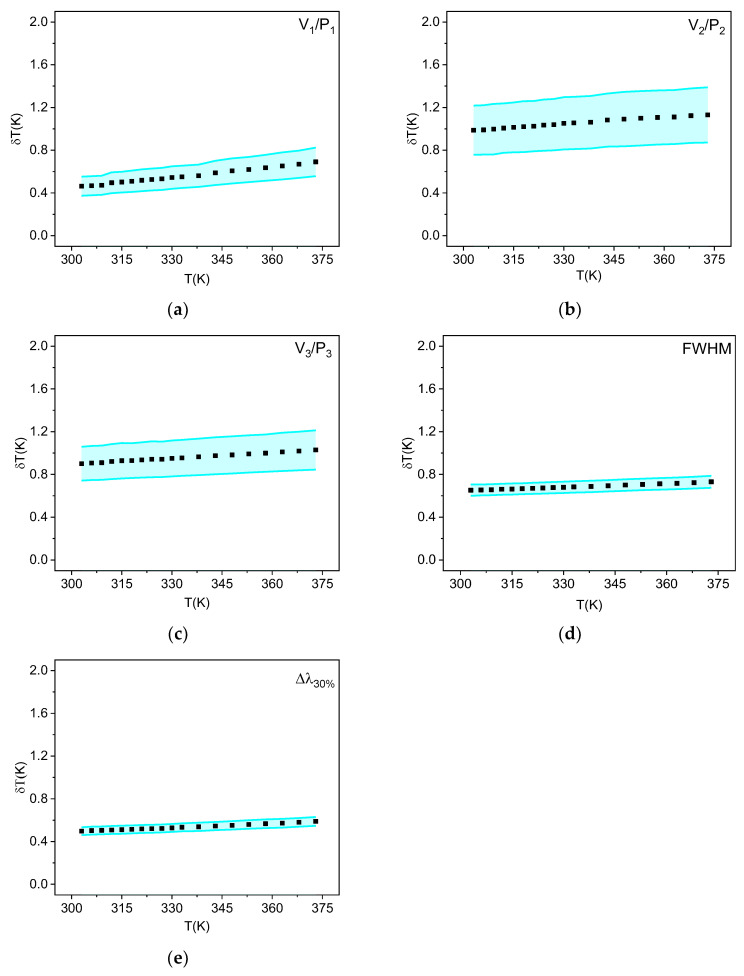
Uncertainty of temperature curves using methods (**a**) V_1_/P_1_; (**b**) V_2_/P_2_; (**c**) V_3_/P_3_; (**d**) FWHM; and (**e**) Δλ_30%_. The error bar of the temperature uncertainty is represented in blue.

**Figure 8 sensors-25-02324-f008:**
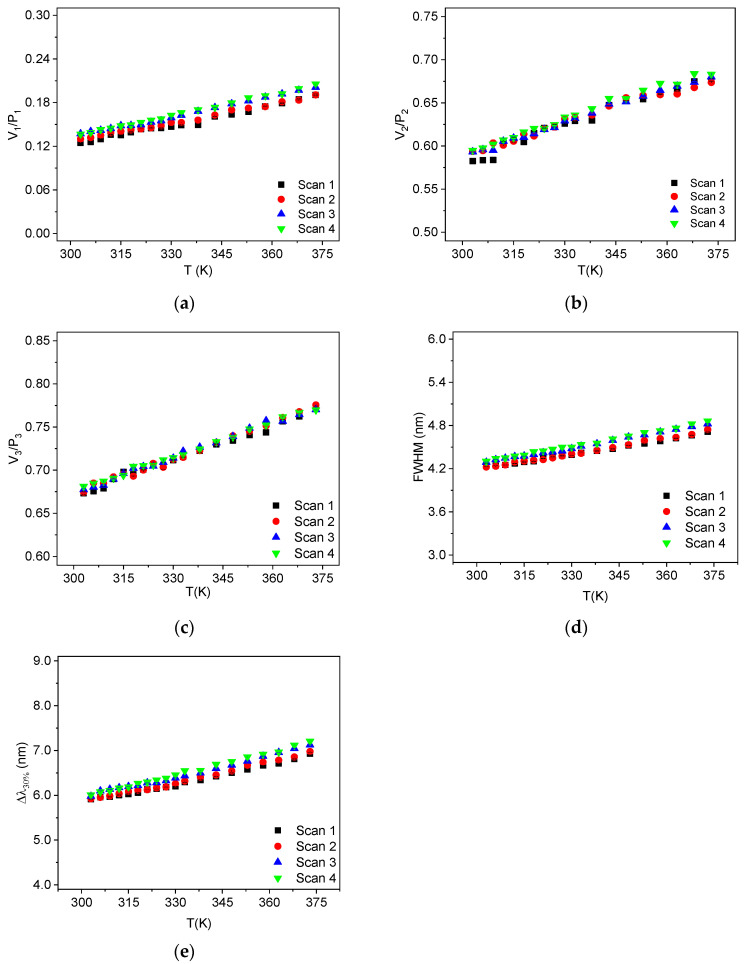
Reproducibility of YF_3_:Nd^3+^ for (**a**) V_1_/P_1_; (**b**) V_2_/P_2_; (**c**) V_3_/P_3_; (**d**) FWHM; and (**e**) Δλ_30%_. Standard deviations were smaller than (**a**) 6.13%, (**b**) 1.51%, (**c**) 0.75%, (**d**) 1.68%, and (**e**) 2.12%.

**Figure 9 sensors-25-02324-f009:**
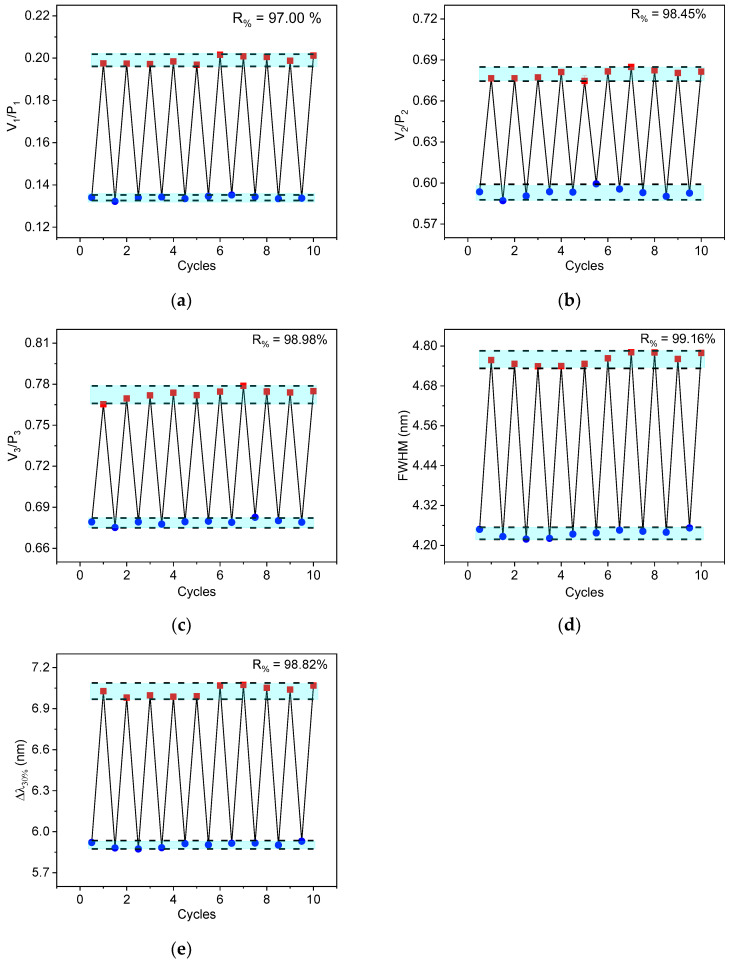
Repeatability curves using method (**a**) V_1_/P_1_; (**b**) V_2_/P_2_; (**c**) V_3_/P_3_; (**d**) FWHM; and (**e**) Δλ_30%_. Thermometric quantities were measured at 303 K (blue) and 373 K (red).

**Table 1 sensors-25-02324-t001:** Comparison of temperature detection performance of materials doped with lanthanide ions using FWHM, VPR, and LIR methodologies.

Material	Ln^3+^	Technique	λ_exc_ (nm)	T (K)	δT (K)	Sr (% K^−1^)	Ref.
Y_2_O_3_	Nd^3+^/Er^3+^	VPR	808	298	4.8	0.45	[34]
Y_2_O_3_	Nd^3+^	VPR	800	294	1.6	0.304	[35]
Y_7_O_6_F_9_	Eu^3+^	VPR	532	318	0.5	0.35	[36]
YVO_4_	Nd^3+^	FWHM	808	298	3	0.14	[31]
YVO_4_	Eu^3+^	FWHM	305	313	3.3	0.10	[32]
CaGdAlO_4_	Tm^3+^/Yb^3+^	FWHM	980	300	1.25	0.124	[33]
YAP	Nd^3+^	FWHM	532	293	0.37	3.3	[22]
Bi_2_SiO_5_	Nd^3+^	LIR	808	310	3.8	0.34	[37]
Bi_4_Si_3_O_12_	Nd^3+^	LIR	745	300	1	0.18	[38]
YNbO_4_	Nd^3+^	LIR	808	303	1.1	0.28	[39]
BiVO_4_	Nd^3+^	LIR	750	310	0.26	1.53	[40]
LaPO_4_	Yb^3+^/Nd^3+^	LIR	980	280	0.02	3.51	[41]
Gd_2_O_3_	Nd^3+^	LIR	532	288	0.14	1.75	[42]
YF_3_	Nd^3+^	LIR	800	303	0.7 ± 0.1	0.22	[26]
YF_3_	Nd^3+^	VPR (V_1_/P_1_)	800	303	0.46 ± 0.09	0.69 ± 0.02	This work
YF_3_	Nd^3+^	VPR (V_2_/P_2_)	800	303	1.0 ± 0.2	0.23 ± 0.02	This work
YF_3_	Nd^3+^	VPR (V_3_/P_3_)	800	303	0.9 ± 0.2	0.194 ± 0.006	This work
YF_3_	Nd^3+^	FWHM	800	303	0.65 ± 0.05	0.170 ± 0.002	This work
YF_3_	Nd^3+^	Δλ_30%_	800	303	0.50 ± 0.03	0.254 ± 0.004	This work

## Data Availability

Dataset available on request from the authors.

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
