# Peer review of "Comparative Analysis of Spectral Broadening Techniques for Optical Temperature Sensing in Yttrium Fluoride (YF3) Doped with Neodymium"

_sensors, 2025, doi:10.3390/s25072324_

Round 1
Reviewer 1 Report
Comments and Suggestions for Authors
This work reports the photoluminescence and optical temperature sensing properties of YF3:Nd3+, which focuses on the emission corresponding to the 4F3/2→4I9/2 transition of Nd3+. The performance of YF3:Nd3+ as an optical temperature sensor was evaluated using the full-width at half-maximum (FWHM), band broadening at 30% of maximum intensity (Δλ30%), and valley-to-peak intensity ratio (VPR) techniques. The results are interesting, but I think this paper can not be accepted before major revisions.
- The manuscript needs to be well organized, for example, Fig.7, 8 needs rearrangement.
- It is difficult to capture the authors, for example, how to obtain the Δλ30%? what is different between FWHM and Δλ30%?
- The legends of Fig. 4c needs to be changed as “The normalized emission spectra of band 4F3/2 → 4I9/2 at different temperatures”. By the way, the emission spectra of band 4F3/2 → 4I9/2 at different temperatures should be given to analysis the photoluminescence properties.
The English could be improved to more clearly express the research.
Author Response
Comment 1: The manuscript needs to be well organized, for example, Fig.7, 8 needs rearrangement.
Response: We thank the referee for this comment. We carefully revised the manuscript and performed some changes that improved the text and figures organization.
Comment 2: It is difficult to capture the authors, for example, how to obtain the Δλ30%? what is different between FWHM and Δλ30%?
Response: We thank the referee for this comment. As we mentioned in the response of the first comment, we have carefully revised the manuscript and believe we could improve the text readability. As requested by the referee, one of the improvements made is regarding the definition of the bandwidth at 30% of the maximum intensity (Δλ30%) and the procedure employed to obtain this value. For this, we have added the following sentences to our manuscript revised version:
i. Figure 3 was modified and now have only two parts: (a) Emission spectrum of YF3:1% Nd3+; (b) Energy level diagram of Nd3+ in the YF3.
ii. A new figure 4 was added, with three parts: (a) The emission spectra of the 4F3/2 → 4I9/2 transition, measured at three different temperatures. (b) The normalized emission spectra of band 4F3/2 → 4I9/2 at different temperatures (c) Peak P1 measured at 303 and 373 K, and the definition of the width measured at half of the maximum intensity (FWHM=λ2-λ1) and measured at 30% of the maximum intensity (Δλ30%=λ4-λ3).
iii. The first two paragraphs of section 3.2 were modified to:
"Figure 3(a) shows the emission spectrum of YF3:1% Nd3+, excited with an 800 nm continuous wave (CW) laser, measured at room temperature. Two emission bands in the infrared region can be observed. One centered at 884 nm, which is assotiated with the radiative decay of electrons from the 4F3/2 level of neodymium to the ground state 4I9/2. The second one is centered around 1056 nm, resulting from the decay from the 4F3/2 level to the excited state 4I11/2. A simplified energy level diagram is presented in Figure 3(b), illustrating the radiative and non-radiative processes of neodymium. In this figure, it can be observed that neodymium electrons are excited by the 800 nm CW laser, transitioning from the ground state 4I9/2 to the higher energy state 4F5/2. From this level, the ions decay non-radiatively to the 4F3/2 level. Subsequently, these ions can decay radiatively either to the ground state, resulting in an emission band located in the first biological window, or to the first excited state 4I11/2, generating an emission band in the second biological window. Moreover, due to the crystaline field of YF3, the energy levels of Nd3+ split into Stark’s sublevels and the emission band is composed by set of narrows emission peaks associtated to the transitions between the sublevels of the 4F3/2 excited state to the sublevels of 4I9/2 and 4I11/2.
To evaluate the potential use of YF3:1% Nd3+ as a temperature sensor, temperature variations were performed to observe the behavior of the material's spectrum. The results are shown in Figure 4(a). In this figure it is presented the emission band at the I-BW, measured at different temperatures, varying from 303 K to 373 K. The overal behavior observed is a typical temperature quenching of the luminescence. Nevertheless, the intensities associated to the individual transitions between the Stark’s sublevels present a more rich behavior. In our previous work [26], the behavior of the 4F3/2 → 4I9/2 emission band of YF3:1% Nd3+ with temperature variations and the transitions of the Stark’s sulevels is detailed. Indeed, exploiting properly the temperature behavior of these emission lines, a LIR-based temperature sensor was proposed and characterized.
To further highlight the luminescent variations of the emission band corresponding to the 4F3/2 → 4I9/2 transition, the data were normalized with respect to peak with the highest intensity, labeled as P1, as shown in Figure 4(b). It is noticeable that all the emission peaks broaden as the temperature increases. To analyze the effect of peak broadening for use in optical temperature sensing, the peak broadening technique was assessed on all peaks. To quantify its width using FWHM, we determine directly from the spectrum the wavelengths at which the corresponding intensities are equal to half of the peak intensity. Hence, the FWHM is obtained simply by the difference between these two wavelengths, as shown in Figure 4(c). Nevertheless, only peak P1 showed sufficient broadening, using FWHM to quantify the spectral width, to allow the use of the band broadening technique. This is observed in detail in Figure 4(c), where the P1 spectra measured at 303 and 373 K are shown.
We also defined the width of the band measured at 30% of the peak intensity, labeled as . This value is also obtained directly from spectrum. First we identify the wavelengths at which the corresponding intensities are equal do 30% of the peak intensity. Then, is the diference between these two wavelength values. In Figure 4(c), it is is also highlighted the width measured at half of the maximum intensity (FWHM) and the width measured at 30% of the peak intensity. It is possible to observe that the broadening at 30% of the maximum intensity of peak P1 is greater than the FWHM broadening. This suggests that performing temperature sensing by monitoring the width at 30% could provide an improvement in the sensitivity obtained in comparison with FWHM method.
Hence, we characterized both widht definifitions as the thermometric parameter of a temperature sensor based on bandwidth. The results are shown in Figure 5(a). As can be seen, both methods exhibit linear behavior and were fitted according to the equation below:..."
Comments 3: The legends of Fig. 4c needs to be changed as “The normalized emission spectra of band 4F3/2 → 4I9/2 at different temperatures”. By the way, the emission spectra of band 4F3/2 → 4I9/2 at different temperatures should be given to analysis the photoluminescence properties.
Response 3: Thanks for this comment. We have performed the required change and add a new figure with the emission spectra of band 4F3/2 → 4I9/2 at different temperatures.
Reviewer 2 Report
Comments and Suggestions for Authors
The authors present a study on the synthesis and characterization of yttrium fluoride (YF3) doped with neodymium (Nd3+), focusing on the optical properties of the obtained microparticles and the impact of different thermometric parameters on relative thermal sensitivity (Sr). While the premise is interesting, this work appears to be a follow-up of a previous publication by the authors (Optical Materials, 2022, 131, 112661). In this regard, no significant material-related novelty is introduced, and even the XRD and TEM data in Figures 2 and 3a of the current submission are identical to Figures 1 and 3b of the earlier publication, respectively.
The main distinction in this study is the introduction of the valley-to-peak intensity ratio (V/P) and bandwidth as thermometric parameters (Δ) to enhance Sr. Although the newly reported Sr values are higher than those obtained using the luminescence intensity ratio approach, it is important to stress that V/P is highly susceptible to background noise fluctuations, as it relies on the intensity at two specific wavelengths. The results in Figure 7 support this concern, showing considerable variations in V/P values at the same temperature. Researchers dealing with luminescence thermometry often use integrated intensity ratios from distinct wavelength ranges to mitigate this issue.
While the current trend in luminescence thermometry emphasizes maximizing Sr, the authors should place greater emphasis on evaluating temperature uncertainty δT=(1/Sr)*(δΔ/Δ)), as it provides a more reliable measure of the material’s thermal resolution based on the relative uncertainty of Δ (δΔ/Δ, related to the signal-to-noise ratio).
Although I appreciate the effort invested in this research, I do not believe it provides enough scientific advancements to the field and therefore cannot recommend the manuscript for publication based on the concerns outlined above.
Author Response
Comments 1: The authors present a study on the synthesis and characterization of yttrium fluoride (YF3) doped with neodymium (Nd3+), focusing on the optical properties of the obtained microparticles and the impact of different thermometric parameters on relative thermal sensitivity (Sr). While the premise is interesting, this work appears to be a follow-up of a previous publication by the authors (Optical Materials, 2022, 131, 112661). In this regard, no significant material-related novelty is introduced, and even the XRD and TEM data in Figures 2 and 3a of the current submission are identical to Figures 1 and 3b of the earlier publication, respectively.
Response: We thank the referee for this comment. Indeed, the referee is correct in the statement that in our previous publication (ref. [26] of the submitted manuscript), we focused on the synthesis and characterization of YF3:Nd3+ particles and evaluated this material for optical thermometry based on luminescence intensity ratio method exploiting transition among Stark’s sublevels. In this regard, the materials synthesis is not the novelty of the present manuscript. Nevertheless, as we are investigation the same material, produced by the same method as in ref. [26], it is expected that the XRD and TEM data are very similar. Hence, we have removed the XRD figure and add a reference to previous work. Moreover, we have replaced the SEM images to illustrate the morphology of the produced particles.
The text of section 2 was also modified to:
The diffraction patterns of the produced undoped YF3 and the YF3:1%Nd3+ samples are essentialy the same reported in our previous work [26]. They indicate that the samples are crystalline with a single phase corresponding to the orthorhombic crystal structure with space group Pnma symmetry [D162h ], matching the ICSD pattern N° 26595.
The SEM images are presented in Figures 2 (a) and (b). In the Figure 2 (a), the images show that the particles have a rod-like shape and are well dispersed. In Figure 2 (b), the magnified image provides a clearer view of how the particles are organized, as well as their shape. The particles have an average size of 644 ± 127 nm in length and 344 ± 47 nm in width.
Comments 2: The main distinction in this study is the introduction of the valley-to-peak intensity ratio (V/P) and bandwidth as thermometric parameters (Δ) to enhance Sr. Although the newly reported Sr values are higher than those obtained using the luminescence intensity ratio approach, it is important to stress that V/P is highly susceptible to background noise fluctuations, as it relies on the intensity at two specific wavelengths. The results in Figure 7 support this concern, showing considerable variations in V/P values at the same temperature. Researchers dealing with luminescence thermometry often use integrated intensity ratios from distinct wavelength ranges to mitigate this issue.
Response: Thank you very much for this comment. Indeed, the main goal of our work was to perform a comparison with different sensing methodologies. Indeed, the reason we have chosen to use YF3:Nd3+ was that its luminescence allowed us to apply different techniques using the same experimental setup. In this sense, we could directly assess the performance in terms of temperature uncertainty, which is not the current trend in luminescence thermometry, as it should. We believe this is now emphasized in the revised version of the manuscript.
The concern regarding the signal to noise ratio of the V/P methodology is analysed in our work. In this sense, I would like to politely disagree with the referee’s statement that the results in Figure 7 should be the only evaluation of the reliability of this method. Indeed, using the data presented in figure 7, we quantified the repeatability, defined in Eq. (6). This quantity is defined as the device's ability to consistently provide the same result in repeated measurements under identical conditions. The obtained results for the V/P method are above 97%, which is a competitive result in comparison to other proposed methodologies. Indeed, the largest variation of V/P values observed in this figure was 0.0047. If you consider that the average value is 0.1986, the fluctuation of the values was smaller than 2.4%.
Moreover, following the referee’s suggestion, we should highlight, in our analysis, the temperature uncertainty. Indeed, as highlighted in the revised version of the manuscript:
“The methodologies based on the bandwidth measured at 30% of the peak (Δλ₃₀%) and the valley-peak intensity ration V1/P1 presented the best results for temperature uncertainty among all tested techniques, varying from 0.50 to 0.68 and from 0.46 to 0.69 K, respectively within the investigated temperature range.” It is remarkable that these values were even better than the obtained valued using Stark’s transitions-based LIR method.
Besides, from the data reproducibility presented in Figure 8 of the manuscript, we could calculate the standard deviation of the thermometric parameters. For all of them, this statistical deviation was smaller than 6.13%.
In summary, the signal to noise ratio is an issue when using V/P technique. However, its influence was carefully analysed and quantified in our work. The obtained results showed that, despite this concern, this method is still competitive in comparison with other methods and materials."
Comments 3: While the current trend in luminescence thermometry emphasizes maximizing Sr, the authors should place greater emphasis on evaluating temperature uncertainty δT=(1/Sr)*(δΔ/Δ)), as it provides a more reliable measure of the material’s thermal resolution based on the relative uncertainty of Δ (δΔ/Δ, related to the signal-to-noise ratio).
Response: Thank you for this remark. We have modified the revised version of the manuscript to highlight this figure of merit. Indeed, we believe that this improved our analysis and the impact of our work. For this, we replaced the 6th and 7th paragraphs of section 3.2 by:
In addition to relative sensitivity, other characteristics of an optical temperature sensor were analyzed: material repeatability, reproducibility and temperature uncertainty [29,30]. These properties are essential for evaluating the sensor's accuracy and reliability. Temperature uncertainty corresponds to the smallest temperature change that can be experimentally detected and was calculated using Equation (5):
δT=(1/Sr)(δQ/Q)
where is the relative uncertainty in determining the thermometric parameter. It is important to note that the temperature uncertainty strongly depends on the experimental setup used, the measured conditions, and the signal-to-noise ratio. The temperature uncertainty can be impoved, for instance, using a system with high relative sensitivity and/or by improving the signal-to-noise ratio of the material's emission spectrum. While te former is achieved with the proper choice of the probe material and thermometric parameter, the later can be done by increasing the integration time or the number of measurementes in the acquisition of the emission spectrum [2,10,28]. Regardless, it is the temperature uncertainty the parameter that provides a more reliable measure of the material’s thermal resolution.
The temperature uncertainties obtained using the band broadening and VPR methods are presented in Figure 7. Except for the temperature uncertainty obtained using the Vâ‚‚/Pâ‚‚ ratio, the uncertainties determined by the other methods yielded excellent results, with values below 1 K within the physiological temperature range (303 – 333 K). This indicates high reliability in temperature measurement, making it highly promising for applications in biological environments. The methodologies based on the bandwidth measured at 30% of the peak (Δλ₃₀%) and the valey-peak intensity ration V1/P1 presented the best results for temperature uncertainty among all tested techniques, varying from from 0.50 to 0.68 and from 0.46 to 0.69 K, respectively within the investigated temperature range. These results are even better than what is obtained using Stark’s transitions-based LIR methodology with the same material and experimental set up [26], which varied from 0.70 to 1.10 K.
Comments 4: Although I appreciate the effort invested in this research, I do not believe it provides enough scientific advancements to the field and therefore cannot recommend the manuscript for publication based on the concerns outlined above.
Response: We hope that after our responses and improvements in the text, you will find that our work provide enough scientific advancements to be accepted for publication.
Reviewer 3 Report
Comments and Suggestions for Authors
The authors have submitted a compelling paper titled "Comparative Analysis of Spectral Broadening Techniques for Optical Temperature Sensing in Yttrium Fluoride (YF₃) Doped with Neodymium." They have successfully synthesized pure-phase Nd-doped YF₃ and provided emission spectra, along with a detailed analysis of various parameters that vary with temperature.
Overall, I like the flow of the paper, as each explanation is followed by corresponding results, effectively maintaining the reader’s attention. The images are clear, well-presented, and appropriately labeled. The table at the end is particularly useful, as it concisely highlights previous work in comparison to the authors’ contributions. The idea and concept of the paper are well-structured and efficiently conveyed within these few pages.
Author Response
Comments: The authors have submitted a compelling paper titled "Comparative Analysis of Spectral Broadening Techniques for Optical Temperature Sensing in Yttrium Fluoride (YF₃) Doped with Neodymium." They have successfully synthesized pure-phase Nd-doped YF₃ and provided emission spectra, along with a detailed analysis of various parameters that vary with temperature.
Overall, I like the flow of the paper, as each explanation is followed by corresponding results, effectively maintaining the reader’s attention. The images are clear, well-presented, and appropriately labelled. The table at the end is particularly useful, as it concisely highlights previous work in comparison to the authors’ contributions. The idea and concept of the paper are well-structured and efficiently conveyed within these few pages.
Response: Thank you for your kind words and encouraging feedback.
Reviewer 4 Report
Comments and Suggestions for Authors
I have no comments on the work, the authors have analyzed the use of YF: Nd as an optical thermometer in great detail and compared the results achieved with other Nd-doped materials. The problem is in Yttrium fluoride Health & Safety information. Signal word: warning, Hazard codes Xn, GHS pictograms Acute toxicity and Health hazard/Hazardous to the ozone layer. Can you comment on the practical application of the YF: Nd optical thermometer?
Author Response
Comments: I have no comments on the work, the authors have analyzed the use of YF: Nd as an optical thermometer in great detail and compared the results achieved with other Nd-doped materials. The problem is in Yttrium fluoride Health & Safety information. Signal word: warning, Hazard codes Xn, GHS pictograms Acute toxicity and Health hazard/Hazardous to the ozone layer. Can you comment on the practical application of the YF: Nd optical thermometer?
Response:
Thank you very much for your comments. Fluorine ions, when isolated and in high concentrations, can be toxic to humans. However, when coordinated with yttrium ions to form yttrium fluoride (YF₃), they become insoluble in biological environment, resulting in a highly stable compound with a significantly reduced risk of toxicity. These properties make YF₃, as well as other rare earth fluorides, of great interest for biomedical applications, including bioimaging and optical sensors. To optimize their biocompatibility and expand their functionalization capacity, encapsulation strategies are often employed, allowing greater control over their interactions with the biological environment and enhancing their applicability in advanced diagnostic and monitoring systems (See, for instance, 10.1117/1.JBO.18.7.076004; doi.org/10.1016/j.msec.2021.111937; 10.1038/srep29746).
Fluorine ions can contribute to ozone depletion in certain gaseous forms such as CFCs, by releasing reactive chlorine ions [https://doi.org/10.1016/j.comptc.2022.113903]. Still, when bonded with yttrium to form yttrium fluoride (YF₃), it becomes a highly stable and insoluble compound. Due to its chemical stability, highly mentioned in the literature, YF₃ does not release reactive fluorine species into the atmosphere, making it environmentally safe and non-harmful to the ozone layer.[10.1016/j.jallcom.2021.162793; doi.org/10.1016/j.optmat.2021.111328; doi.org/10.1016/j.optmat.2023.113839; doi.org/10.1016/j.optmat.2019.04.050].
Round 2
Reviewer 1 Report
Comments and Suggestions for Authors
The manuscript can be accepted in present form.
Reviewer 2 Report
Comments and Suggestions for Authors
The authors have properly addressed my criticisms, improving the overall quality of the manuscript and making it more suitable for publication in this journal. However, I would like to highlight two minor points that can be addressed during the proof revision, eliminating the need for an additional review round:
i) Since the new version of Figure 2b is simply an amplification of Figure 2a, it does not provide additional information regarding size or morphology. Then, I suggest using Figure 2b from the revised manuscript as Figure 2a and reinstating Figure 2b from the original manuscript (particle size distribution) to give the readers more information.
ii) The symbols in Equation 5 and the corresponding text are missing.